

# Global scale evaluation of precipitation datasets for hydrological modelling

Solomon H. Gebrechorkos[1,2], Julian Leyland[2], Simon J. Dadson[1], Sagy Cohen[3], Louise Slater[1], Michel Wortmann[1], Philip J. Ashworth[4], Georgina L. Bennett[5], Richard Boothroyd[6], Hannah Cloke[7,8], Pauline Delorme[9], Helen Griffith[7], Richard Hardy[10], Laurence Hawker[11], Stuart McLelland[9], Jeffrey Neal[11], Andrew Nicholas[5], Andrew J. Tatem[2], Ellie Vahidi[5], Yinxue Liu[1], Justin Sheffield[2], Daniel R. Parsons[10], Stephen E. Darby[2]

[1]School of Geography and the Environment, University of Oxford, Oxford, UK
[2]School of Geography and Environmental Science, University of Southampton, Southampton, SO17 1BJ, United Kingdom
[3]Department of Geography and the Environment, University of Alabama, Tuscaloosa, AL, USA
[4]School of Applied Sciences, University of Brighton, Sussex, BN2 4AT
[5]Department of Geography, Faculty of Environment, Science and Economy, University of Exeter, Exeter, EX4 4RJ, United Kingdom
[6]School of Geographical & Earth Sciences, University of Glasgow, UK
[7]Department of Geography and Environmental Science, University of Reading, UK
[8]Department of Meteorology, University of Reading, UK
[9]Energy and Environment Institute, University of Hull, Hull, United Kingdom
[10]Department of Geography, Durham University, Lower Mountjoy, South Road, Durham, DH1 3LE
[11]School of Geographical Sciences, University of Bristol, Bristol, BS8 1SS, UK

*Correspondence to*:: Solomon H. Gebrechorkos (solomon.gebrechorkos@ouce.ox.ac.uk)





**Abstract.** Precipitation is the most important driver of the hydrological cycle but is challenging to estimate over large scales from satellites and models. Here, we assessed the performance of six global and quasi-global high-resolution precipitation datasets (ERA5 global reanalysis (ERA5), Climate Hazards group Infrared Precipitation with Stations version 2.0 (CHIRPS), Multi-Source Weighted-Ensemble Precipitation version 2.80 (MSWEP), TerraClimate (TERRA), Climate Prediction Centre Unified version 1.0 (CPCU) and Precipitation Estimation from Remotely Sensed Information using Artificial Neural Networks-Cloud Classification System-Climate Data Record (PERCCDR)) for hydrological modelling globally and quasi-globally. We forced the WBMsed global hydrological model with the precipitation datasets to simulate river discharge from 1983 to 2019 and evaluated the predicted discharge against more than 1800 hydrological stations worldwide, using a range of statistical methods. The results show large differences in the accuracy of discharge predictions when using different precipitation input datasets. Based on evaluation at annual, monthly and daily time scales, MSWEP followed by ERA5 demonstrated a higher CC and KGE than other datasets for more than 50% of the stations. Whilst, ERA5 was the second-highest performing dataset, it showed the highest error and bias in about 20% of the stations. The PERCCDR is the least well-performing dataset with large bias (percentage of bias up to 99%) and errors (normalised root mean square error up to 247%) with a higher KGE and CC than the other products in less than 10% of the stations. Even though MSWEP provided the highest performance overall, our analysis reveals high spatial variability, meaning that it is important to consider other datasets in areas where MSWEP showed a lower performance. The results of this study provide guidance on the selection of precipitation datasets for modelling river discharge for a basin, region or climatic zone as there is no single best precipitation dataset globally. Finally, the large discrepancy in the performance of the datasets in different parts of the world highlights the need to improve global precipitation data products.


## 1. Introduction

Whilst precipitation is one the most important components of the global hydrological cycle and regulates the climate system (Miao et al., 2019; Sadeghi et al., 2021), it remains one of the most challenging variables to estimate at a global scale using satellite data and modelling approaches (Michaelides et al., 2009; Kidd and Levizzani, 2011; Beck et al., 2017a; Ursulak and Coulibaly, 2021). Reliable precipitation data with sufficient spatial and temporal coverage and accurate representation of extreme events is crucial for developing water resource management and planning strategies, hydrological applications including forecasting hydrological extremes, and climate change analysis (Mehran and AghaKouchak, 2014; Nguyen et al., 2018; Sadeghi et al., 2021; Acharya et al., 2019). Observed precipitation from meteorological stations is typically used at local to river basin scale with gauge-based gridded precipitation datasets, such as from the Global Historical Climatology Network (Menne et al., 2012), developed to study climate and hydrology over larger scales. However, precipitation from gauges and gauge-based gridded datasets have several drawbacks such as limited spatial and temporal coverage, prevalence of missing values, and limited accuracy in sparsely populated and remote areas (Kidd and Levizzani, 2011; Reichle et al., 2011; Kidd et al., 2017; Sun et al., 2018; Gebrechorkos et al., 2018; Hafizi and Sorman, 2022). In addition, data-sharing policies have caused significant challenges in obtaining data, particularly in developing countries (Gebrechorkos et al., 2018; Hafizi and Sorman, 2022).

Over the last few decades, several global and quasi-global precipitation datasets have been developed that address some of these challenges and can be used to drive hydrological models at regional and global scales. The precipitation datasets differ in terms of their spatial resolution, spatial coverage (e.g., global or regional), data sources (e.g., gauge, satellite, reanalysis, and radar), temporal resolution (e.g., sub-daily and daily), and length of record. It is therefore important to evaluate the accuracy of the datasets before they are used to drive global or regional scale hydrological models. Most studies have evaluated precipitation datasets using observed data from field-based meteorological stations at a range of scales (e.g., Beck et al., 2017a; Gebrechorkos et al., 2018; Xiang et al., 2021; Sun et al., 2018; Hong et al., 2022; Wati et al., 2022; AL-Falahi et al., 2020; Ahmed et al., 2019; Fallah et al., 2020). Hydrological models have also been used to assess the quality of the precipitation dataset by comparing simulated and observed discharge across different spatial scales (e.g., Mazzoleni et al., 2019; Beck et al., 2017a; Zhu et al., 2018; Raimonet et al., 2017; Guo et al., 2018; Wang et al., 2020; Salehi et al., 2022; Zhu et al., 2018; Seyyedi et al., 2015). In principle, this latter approach is able to identify the precipitation datasets which best represent hydrological variability including extremes, even in catchments where there have been multiple drivers of change.

The are a limited number of studies assessing multiple precipitation datasets for global hydrological model applications (Voisin et al., 2008; Beck et al., 2017a; Mazzoleni et al., 2019). Beck et al., (2017a) compared the performance of multiple precipitation datasets (e.g., the Climate Hazards group Infrared Precipitation with Stations (CHIRPS, version 2.0), Multi-Source Weighted-Ensemble Precipitation (MSWEP, version 2.0), European Centre for Medium-range Weather Forecasts ReAnalysis Interim (ERA-Interim), and National Centers for Environmental Prediction Climate Forecast System Reanalysis (NCEP-CFSR)) for global hydrological modelling. Mazzoleni et al. (2019) evaluated multiple precipitation datasets including MSWEP (Version 2.1) and CHIRPS in eight river basins on different continents. Both Beck et al. (2017a) and Mazzoleni et al. (2019) found





that merged satellite-observation precipitation products showed the best performance compared to satellite-only
products. These studies exclusively concentrate on a daily time scale, evaluating performance solely through the
Nash-Sutcliffe Efficiency (NSE). Neither study extends this assessment to monthly and annual time scales, and
notably, they do not assess the hydrological extremes which are often considered important to capture. Here, we
build upon the work by Beck et al., (2017a) by adding recently developed high-resolution precipitation datasets
such as the ERA5 (Hersbach et al., 2020), TerraClimate (Abatzoglou et al., 2018) and Precipitation Estimation
from Remotely Sensed Information using Artificial Neural Networks-Cloud Classification System-Climate Data
Record (PERSIANN-CCS-CDR, Sadeghi et al., 2021) and the latest MSWEP version (2.80). These additions
significantly broaden the scope of our study, offering a diverse range of products with distinct methodologies. In
addition, we use multiple statistical metrics to evaluate the performance of the precipitation products for
hydrological modelling at daily, monthly and annual time scales and for daily extremes, which represents a current
gap in the modelling literature.
The aim of this study is to undertake a comprehensive evaluation, spanning various temporal and spatial scales,
to examine how different input precipitation datasets impact the predictions of a global hydrological model. We
assess six high-resolution precipitation datasets, each with records spanning over 30 years. A comprehensive and
physically based gridded global hydrological model (WBMsed; Cohen et al., (2013)) is used to simulate river
discharge globally. The modelled discharge, derived from the six precipitation datasets, is assessed across the
various time scales by comparing it with observed discharge data collected from 1825 river gauge stations
worldwide. Furthermore, we assess the performance of the precipitation products by examining their accuracy in
representing daily extreme precipitation events across various percentiles. In summary, this research offers a
thorough evaluation of this set of diverse precipitation products, spanning from daily extreme events to annual
time scales, providing an invaluable resource for selecting appropriate basin-to-regional-to-global scale inputs for
hydrological modelling applications.

### 2. Data and methods

In the following sections, we outline the various input and evaluation datasets which were used within the
WBMsed hydrological modelling framework. The statistical evaluation methods used to assess the results are also
outlined.

### 2.1. Input global and quasi-global precipitation datasets

The precipitation datasets used herein are selected based on their length of record (>30 years period), and spatial
coverage (global and quasi-global) (Table 1). The selected precipitation datasets are the ERA5 global reanalysis
(ERA5), Climate Hazards group Infrared Precipitation with Stations version 2.0 (CHIRPS), Multi-Source
Weighted-Ensemble Precipitation version 2.80 (MSWEP), TerraClimate (TERRA), Climate Prediction Centre
Unified version 1.0 (CPCU) and Precipitation Estimation from Remotely Sensed Information using Artificial
Neural Networks-Cloud Classification System-Climate Data Record (PERCCDR). Due to their spatial coverage,
CHIRPS and PERCCDR are evaluated only up to latitudes of 50°N and 60°N, respectively (Table 1). Each dataset
was subsequently used to force the WBMsed hydrological model, to generate streamflow estimates.


ERA5 is the fifth generation European Centre for Medium-Range Weather Forecasts (ECMWF) reanalysis data
available globally from 1940 to present (Hersbach et al., 2020). ERA5 combines modelled data and observations
to create a complete and consistent global climate dataset using data assimilation methods. ERA5 provides
improved precipitation representation such as the inclusion of tropical cyclones when compared to the ERA-
Interim (He et al., 2020; Jiao et al., 2021). ERA5-Land is available at higher spatial resolution (0.1°) from 1950
to present compared to ERA5 (Hersbach et al., 2020). The data is freely available from Copernicus Climate Data
Store (https://cds.climate.copernicus.eu/cdsapp#!/dataset/reanalysis-era5-land?tab=overview).
CHIRPS is a high-resolution quasi-global rainfall product primarily developed for monitoring droughts and global
environmental changes (Funk et al., 2015). CHIRPS provides coupled gauge-satellite precipitation estimates with
a 0.05° spatial resolution and long-period records. The product is developed by combining satellite-only Climate
Hazards group Infrared Precipitation (CHIRP), Climate Hazards group Precipitation climatology (CHPclim), and
data from ground stations. CHIRP and CHPclim were developed based on calibrated infrared cold cloud duration
(CCD) precipitation estimates and ground station data from the Global Historical Climate Network (GHCN). The
product is available at the Climate Hazards Group (https://www.chc.ucsb.edu/data/chirps/) on daily, 10-day, and
monthly timescales from the 1981-near present. Due to its availability at high spatial and temporal resolution,
CHIRPS is widely used in hydrological studies (Luo et al., 2019; Gebrechorkos et al., 2020; Geleta and Deressa,
2021; Wang et al., 2021; Opere et al., 2022; Day and Howarth, 2019; Gebrechorkos et al., 2019) and modelling
of hydrological extremes such as droughts and floods (Chen et al., 2020; Mianabadi et al., 2022; Peng et al., 2020).
MSWEP is a global high-resolution (0.1°) precipitation product developed by merging multiple datasets such as
ground stations (~77,000), satellite-based rainfall estimates, and reanalysis data (Beck et al., 2019b). MSWEP
includes station data from the Global Historical Climatology Network-Daily (GHCN-D), Global Summary of the
Day (GSOD), Global Precipitation Climatology Centre (GPCC), and WorldClim; satellite data from the Global
Satellite Mapping of Precipitation (GSMaP), Tropical Rainfall Measuring Mission (TRMM) Multi-satellite
Precipitation Analysis (TMPA-3B42RT), Climate Prediction Center morphing technique (CMORPH), and
Gridded Satellite (GridSat); and reanalysis datasets such as the Japanese 55-year Reanalysis (JRA-55) and
European Centre for Medium-Range Weather Forecasts (ECMWF) interim reanalysis (ERA-Interim) (Beck et al.,
2017b, 2019b). MSWEP has been widely used in regional and global scale hydrological studies such as for floods
and droughts (Gu et al., 2023; Gebrechorkos et al., 2022b; Reis et al., 2022; Wu et al., 2018; Sun et al., 2022;
Gebrechorkos et al., 2022c; Xiang et al., 2021; López López et al., 2017) and for developing high-resolution
global scale hydrological extreme and climate datasets and regional drought monitoring (Gebrechorkos et al.,
2023, 2022a; Li et al., 2022b). MSWEP is available from 1979-present at multiple timescales (e.g., 3 hourly) and
can be accessed from the GloH2O website (https://www.gloh2o.org/mswep/).
TerraClimate (TERRA) is a high-resolution (0.04°) terrestrial monthly climate (e.g., precipitation and
temperature) and climatic water-balance dataset available from 1958-2020 (Abatzoglou et al., 2018). TERRA was
developed by combining high and coarse spatial resolution datasets such as WorldClim climatological normals
and Climatic Research Unit gridded Time Series (CRU TS) and JRA-55, respectively. The data was evaluated
against ground observation from the Historical Climate Network and exhibited better performance than the CRU-





TS (Abatzoglou et al., 2018). The monthly climate and climatic water balance is available from the Climatology
Lab website (https://www.climatologylab.org/).
CPCU is a gauge-based analysis of daily precipitation datasets available globally from 1979 to present (Chen et
al., 2008). CPCU is the product of the CPC Unified Precipitation project at NOAA Climate Prediction Center.
The product uses data from more than 30,000 (1979-2005) and 17,000 (2006-present) stations. The CPCU data is
publicly       available       at       the       NOAA       Physical       Sciences       Laboratory       (PSL,
https://downloads.psl.noaa.gov/Datasets/cpc_global_precip/) and has been used for hydrological and climate
studies (Beck et al., 2017a; Zhu et al., 2021; Hou et al., 2014).
The PERCCDR is a quasi-global (latitude from 60°S to 60°N) dataset developed at the University of California
(Sadeghi et al., 2021). PERCCDR provides precipitation estimates at high spatial (0.04°) and temporal (3-hourly)
resolutions from 1983 to present. The dataset is developed using the rain rate output from the PERSIANN-CCS
model, which uses GridSat-B1 IR and NOAA Climate Prediction Center (CPC-4km) IR data. Compared to other
PERSIANN precipitation datasets, PERCCDR provides a realistic representation of precipitation extremes
globally and shows better agreement with CPCU precipitation (Sadeghi et al., 2021). The PERCCDR has been
used in hydrological studies (Salehi et al., 2022; Eini et al., 2022) and is freely available from the Center for
Hydrometeorology and Remote Sensing (CHRS) Data Portal (https://chrsdata.eng.uci.edu/).
Table 1. The six precipitation datasets used in this study, their spatial and temporal resolution, spatial coverage
and data sources.

| Abbreviation | Full name | Spatial resolution and coverage | Temporal resolution | Temporal coverage | Data source | Reference |
|---|---|---|---|---|---|---|
| ERA5 | ECMWF (European Centre for Medium-Range Weather Forecasts) Reanalysis V5 | 0.1°, global | Sub-daily | 1979-present | Gauge and reanalysis | (Hersbach et al., 2020) |
| CHIRPS | Climate Hazards group Infrared Precipitation with Stations (CHIRPS) version 2.0 | 0.05°, quasi global (50°S-50°N) | Daily | 1981-present | Gauge, satellite, and reanalysis | (Funk et al., 2015) |
| MSWEP | Multi-Source Weighted-Ensemble Precipitation (MSWEP) version 2.80 | 0.1°, global | Daily | 1979-present | Gauge, satellite, and reanalysis | (Beck et al., 2019b) |



| TERRA | TerraClimate | 0.042°, global | Monthly | 1958-present | Gauge and reanalysis | (Abatzoglou et al., 2018) |
|---|---|---|---|---|---|---|
| CPCU | Climate Prediction Centre (CPC) Unified V1.0 | 0.5°, global | Daily | 1979-present | Gauge only | (Chen et al., 2008) |
| PERCCDR | Precipitation Estimation from Remotely Sensed Information using Artificial Neural Networks-Cloud Classification System-Climate Data Record (PERSIANN-CCS-CDR) | 0.04°, Quasi global (60°S-60°N) | Sub-daily | 1983-present | Gauge and satellite | (Sadeghi et al., 2021) |

**2.2. WBMsed hydrological model**

The WBMsed (Cohen et al., 2013, 2014) hydrological model is used to assess the performance of the different precipitation datasets for hydrological modelling globally. WBMsed is a global-scale hydrogeomorphic model, an extension of the WBMplus global hydrology model (Wisser et al., 2010), which is part of the FrAMES biogeochemical modelling framework (Wollheim et al., 2008). The WBMplus model is one of the first Global Hydrological Models (GHMs) applied to a global domain (Cohen et al., 2013; Grogan et al., 2022). The model represents the major hydrological cycle components of the land surface and tracks the balances and fluxes between the atmosphere, surface water storages, vegetation, runoff, and groundwater (Grogan et al., 2022). The model includes hydrological infrastructure (e.g., dams), agricultural water requirements, and domestic and industrial water uses. A high-resolution gridded river network connects grid cells, which allows the routing of fluxes downstream (e.g., streamflow). The model requires several climate datasets as input in addition to precipitation, including temperature, humidity, air pressure and wind speed. We use an identical model setup to that used by Cohen et al., (2022) with all input datasets as detailed in Cohen et al. (2013) with updates to air temperature which used the daily ERA5 (Hersbach et al., 2020) dataset re-gridded at 10 arc-minutes resolution; reservoir capacity—global reservoir and dam database (GRanD v1.3; Lehner et al., (2011)); and flow network—6 arc-minute HydroSTN30 network which is a derivative from HydroSHEDS high resolution gridded network (Lehner et al., 2008). In addition, we used each of our six input precipitation datasets, ERA5, CHIRPS, MSWEP, TERRA, CPCU, and PERCCDR in turn, keeping all other parameters and inputs the same. Even though WBMsed can disaggregate monthly time series into daily, TERRA (only available at monthly resolution, see table 1) is evaluated on monthly and annual time scales, whilst all other datasets are evaluated at daily time scales in addition. WBMsed simulations were run at 0.1° (~11km at the equator) spatial and daily temporal resolutions. Several WBMsed streamflow validation analyses have been reported previously (e.g., Cohen et al., 2022; Dunn et al., 2019; Cohen et al., 2014, 2013; Moragoda and Cohen, 2020), which indicate that the model represents the long-term average observed streamflow globally. Cohen et al. (2022) report $R^2$=0.99 in 30-year average prediction against USGS gauge data and a global river dataset.





**2.3. Observed river discharge from ground stations**

Observed daily and monthly river discharge used to evaluate the hydrological model were obtained from the Global Runoff Data Centre (GRDC, 2023). The GRDC is an international data archive (https://www.bafg.de/GRDC/), which hosts data for over 10,000 hydrological stations. The number of stations with a length of record greater than 10 years during the evaluation period (1981-2019) are limited. Due to the spatial resolution of the input datasets and the model simulations (~11x11 km), we only consider stations with a catchment area of greater than 100 km$^2$. Overall, 1825 suitable stations were identified with daily and monthly records, largely in North and South America, Europe and Australia, with very few stations in Africa and Asia (Figure 1).

**2.4. Evaluation metrics**

Several methods are used to assess the modelled discharge using the streamflow observations: the Pearson correlation coefficient (CC, Eq. 1), Kling-Gupta Efficiency (KGE, Eq. 2) (Gupta et al., 2009), Root-Mean-Square Error (RMSE, Eq.3) and Percentage of bias (Pbias, Eq.4). A KGE value of 1.0 indicates a perfect match between the observed and simulated discharge, whereas values lower than -0.41 show that the model is worse than using the mean of the observed discharge as a predictor (Knoben et al., 2019). For spatial comparison, the RMSE is normalised by the standard deviation of the observed data (NRMSE; Eq. 5).

$$CC = \frac{\sum_{i=1}^{N}(M_i - \bar{M}) * (O_i - \bar{O})}{\sqrt{\sum_{i=1}^{N}(M_i - \bar{M})^2} * \sqrt{\sum_{i=1}^{N}(O_i - \bar{O})^2}} \tag{1}$$

$$KGE = 1 - \sqrt{(r-1)^2 + (\alpha - 1)^2 + (\beta - 1)^2} \tag{2}$$

$$RMSE = \sqrt{\frac{\sum_{i=1}^{N}(O_i - M_i)^2}{N}} \tag{3}$$

$$Pbias = \frac{\sum_{i=1}^{N}(M_i - O_i)}{\sum_{i=1}^{N}O_i} * 100 \tag{4}$$

$$NRMSE = \frac{RMSE}{SD} * 100 \tag{5}$$

where r is the linear correlation between observed (O) and modelled (M) discharge and α and β are the variability and bias ratios, respectively. The NRMSE and SD are the normalised RMSE and standard deviation, respectively. To assess the performance of the precipitation datasets for representing hydrological extremes, the 90th (Q10) and 10th (Q90) percentile are used, which indicates high and low flows, respectively. The Q10 and Q90 represent the streamflow value that is equalled or exceeded 10% and 90% of the time, respectively.





### 3. Results

The performance of the six different precipitation datasets in simulating discharge is evaluated at annual, monthly and daily time steps and for extremes during the period 1983-2019. The WBMsed output discharge forced by the six precipitation datasets is referred to as ERA5, CHIRPS, MSWEP, TERRA, CPCU, and PERCCDR below.

### 3.1. Performance of the six precipitation datasets for annual discharge prediction

The temporal correlation coefficient (CC) between the observed and simulated annual discharge based on the six precipitation datasets is summarised in Figure 1. Most of the datasets, particularly ERA5, MSWEP, and CHIRPS, showed a high CC in basins of Europe (e.g., Danube basin), South America (e.g., Rio de la Plata-Parana), North America and Australia (e.g., Murray-Darling). MSWEP and ERA5 showed the highest CC for 34% and 32% of the stations, respectively, followed by CPCU and CHIRPS. The TERRA and PERCCDR were the least well-performing datasets with lower CC overall, and a higher CC than other datasets for less than 9% of stations. The median CC of MSWEP and ERA5 is 0.82 and 0.8, respectively. MSWEP and TERRA showed lower Pbias and NRMSE compared to the other datasets (Figures S1 and S2). ERA5 and PERCCDR showed a high NRMSE (up to 247%) and Pbias (up to 99%) for more than 46% of stations. Similar to the CC, ERA5 and MSWEP outperformed the other datasets for KGE, with higher values for 32% and 27% of stations, respectively. The performance of MSWEP and ERA5 is higher in basins of Europe, South America, and Australia compared to Asia and Africa. The median KGE values of ERA5 and MSWEP are 0.33 and 0.32, respectively (Figure 2). The PERCCDR and CPU demonstrate high KGE only in about 9% of the stations, with median values of 0.10 and 0.13, respectively. Based on the annual CC and KGE, there is no single precipitation dataset that is best everywhere, and even the least well-performing dataset overall shows better performance in some stations (Figure 3). Figure 3 summarizes the spatial representation of precipitation dataset performance, highlighting the individual datasets exhibiting the highest CC and KGE values at each observation point.



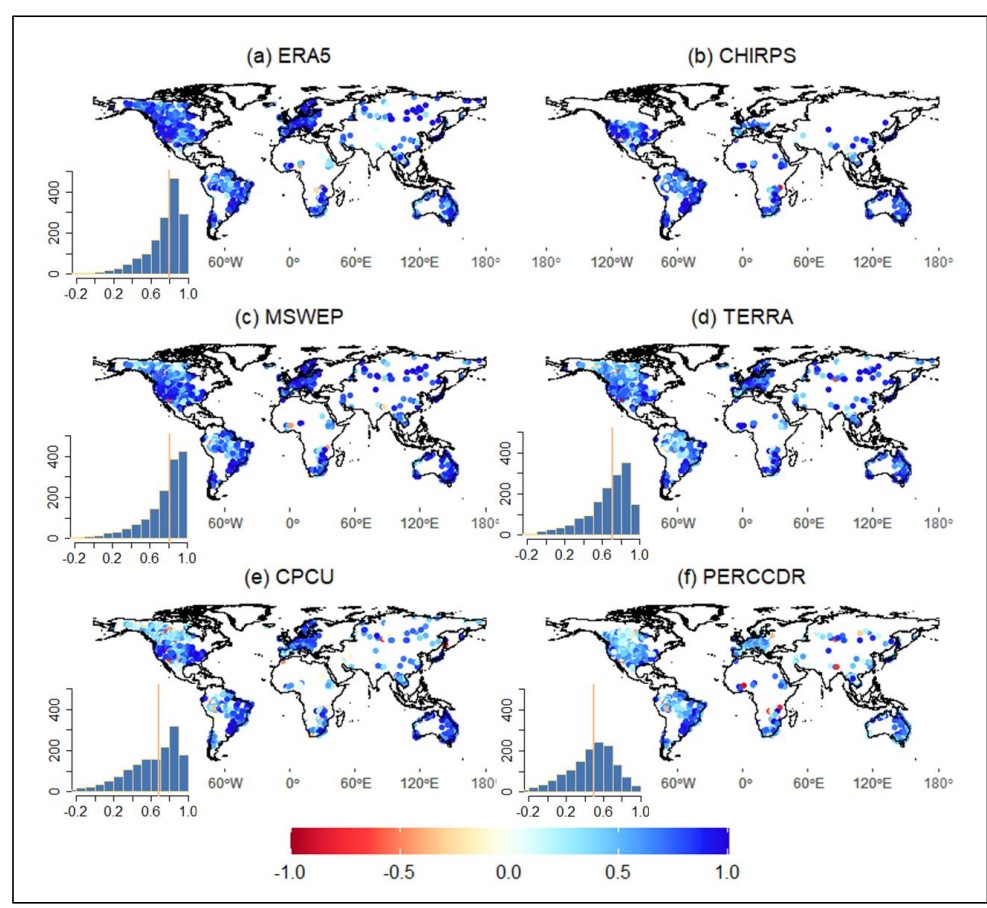

258

**Figure 1: Correlation (CC) between annual observed and modelled streamflow data using a) ERA5, b) CHIRPS, c)
MSWEP, d) TERRA, e) CPCU and f) PERCCDR precipitation datasets. The inset histograms show the frequency
distribution of the monthly CC, with the yellow vertical line indicating the median value.**

262

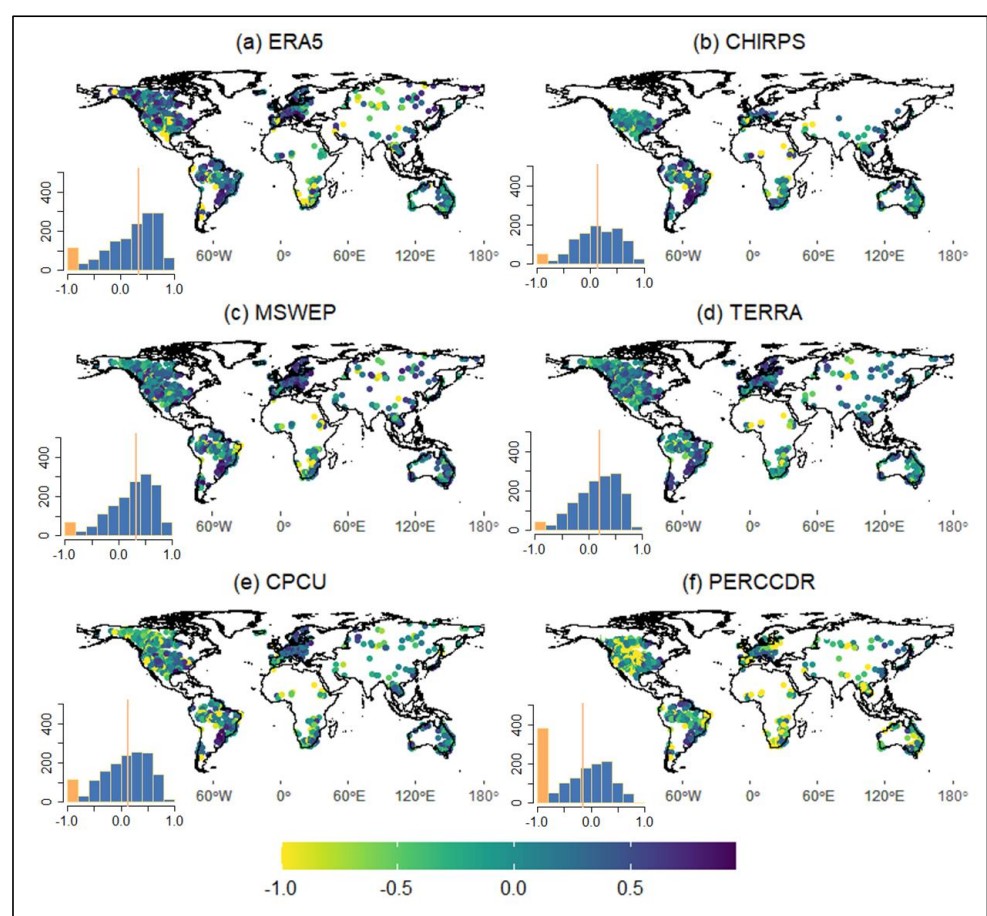

**Figure 2: KGE between observed and modelled annual streamflow based on a) ERA5, b) CHIRPS, c) MSWEP, d) TERRA, e) CPCU, and f) PERCCDR precipitation datasets. KGE values below -0.41 indicate bad model performance than using observed discharge mean as a predictor. The inset histograms show the frequency distribution of the monthly KGE. KGE values lower than -1 are highlighted in yellow. The yellow vertical line indicates the median value.**

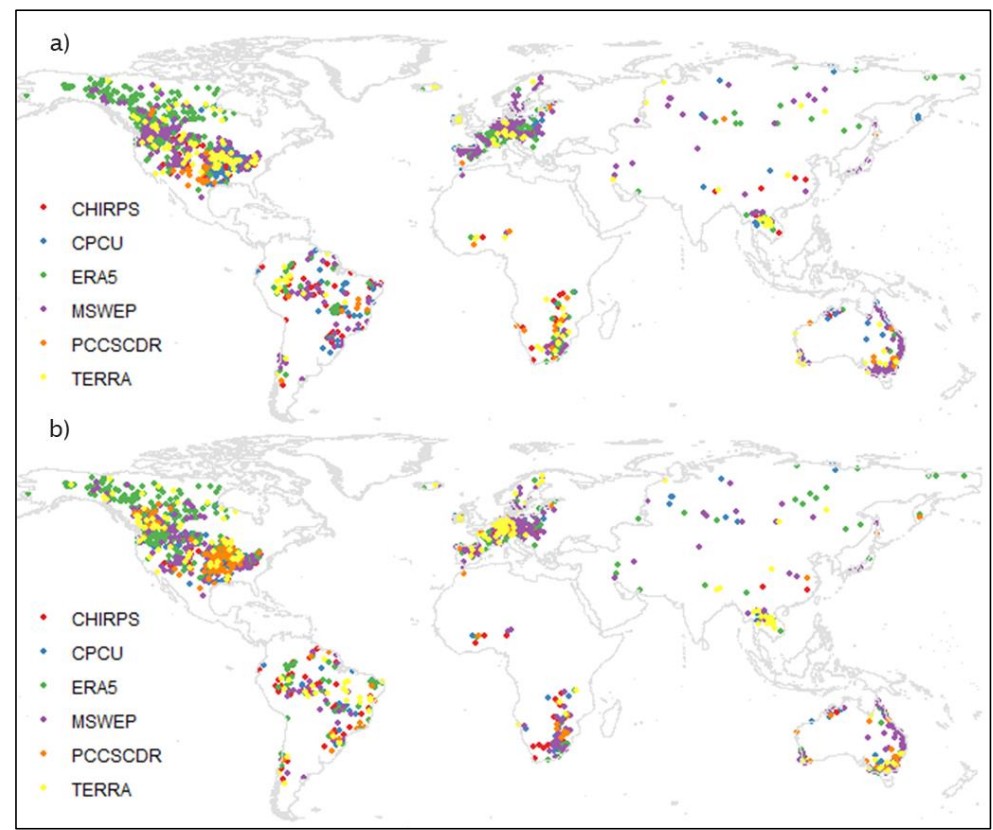

**Figure 3: The best performing precipitation dataset (ERA5, CHIRPS, MSWEP, TERRA, CPCU, and PERCCDR) at each of the observed discharge stations based on annual CC (a) and KGE (b).**

### 3.2. Performance of the six precipitation datasets for monthly discharge predictions

The six precipitation datasets consistently demonstrate high CC at a monthly scale in large parts of the world, except in some rivers of Canada and Australia (Figure 4). The monthly CC, similar to the annual CC, shows a relatively better performance of MSWEP with a median CC of 0.76. TERRA is the second-best with a median CC of 0.69. MSWEP and TERRA show a higher CC than other datasets in 35% and 28% of the stations, respectively. ERA5 and CHIRPS are ranked as the third and fourth datasets with a median CC of 0.71 and 0.75, respectively. CPCU and PERCCDR are the least well-performing datasets, which only show the highest CC in less than 6% of the stations with a median CC of 0.67 and 0.56, respectively.

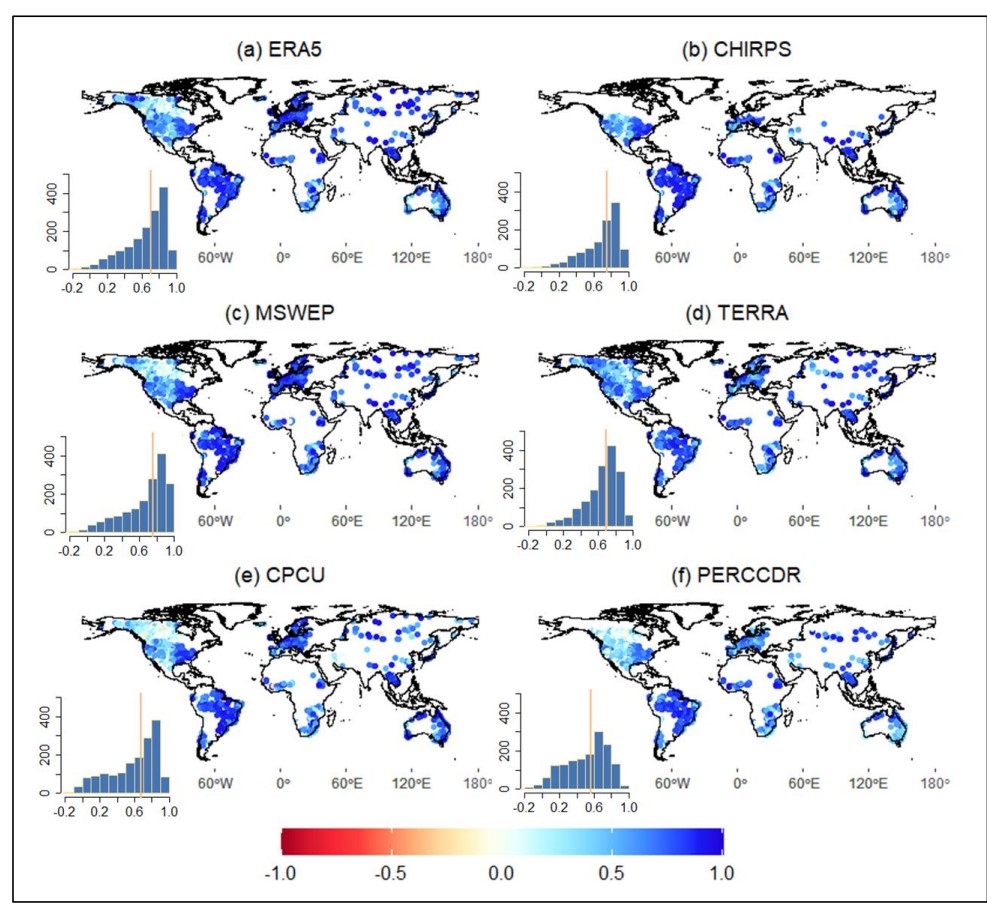


**Figure 4: Correlation (CC) between monthly observed and modelled streamflow data based on a) ERA5, b) CHIRPS, c) MSWEP, d) TERRA, e) CPCU and f) PERCCDR precipitation datasets. The inset histograms show the frequency distribution of the monthly CC, with the yellow vertical line indicating the median value.**

The monthly KGE also indicates the better performance of ERA5 and MSWEP for 26% and 24% of stations,
respectively (Figure 5). MSWEP showed a lower Pbias and NRMSE than all datasets, except in 5% of the stations
(Figures S3 and S4). Compared to MSWEP,  ERA5 showed a larger Pbias and NRMSE in 15% and 19% of the
stations. TERRA, a third-best performing dataset based on KGE (18% of stations), shows a lower monthly Pbias
and RMSE in 85% of the stations compared to CHIRPS, ERA5, and PERCCDR. Compared to all datasets, the
PERCCDR showed a higher NRMSE and Pbias in 55% and 28% of the stations, respectively. The spatial
representation of precipitation dataset performance is summarized in Figure S5, highlighting the regions where
individual datasets demonstrate higher monthly CC and KGE values.

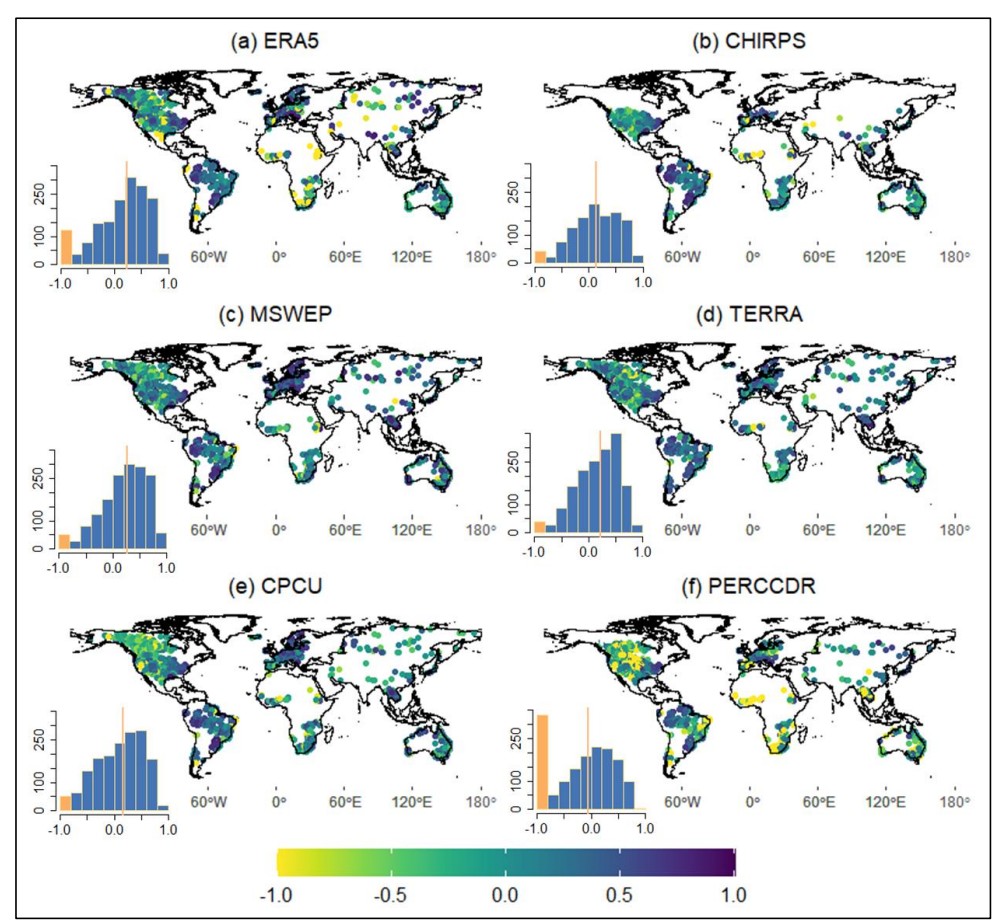

**Figure 5: Monthly KGE values between observed and modelled streamflow based on a) ERA5, b) CHIRPS, c) MSWEP,**
**d) TERRA, e) CPCU and f) PERCCDR precipitation datasets. KGE values below -0.41 indicate model performance**
**that is worse than using the observed discharge mean as a predictor. The inset histograms show the frequency**
**distribution of the monthly KGE. KGE values lower than -1 are highlighted in yellow. The yellow vertical line indicate**
**the median value.**

Figure 6 shows the time series of monthly observed and modelled streamflow based on the six precipitation
datasets for selected locations in basins of Africa (Niger, Lokoja), Asia (Mekong, Khong-Chiam), South America
(Amazon, Missao-Icana), North America (Mississippi, Savannah), Australia (North East Coast, Mirani-Weir),
and Europe (Danube, Dunaalmas). The basins were chosen to represent a good range of climatic regions and
drainage areas where there was availability of a long time series of observed data. In Niger, the observed monthly
flow and variability at Lokoja station are very well reproduced by CHIRPS and TERRA with a CC of 0.88 and
0.85, respectively (Figure 6a). Even though CPCU showed a lower CC (0.64) at Lokoja, it showed a higher KGE
(0.62) and lower Pbias (0.4%) compared to the other products. At Lokoja, PERCCDR is the least well-performing
dataset with the highest RMSE and Pbias and lowest KGE. The monthly variability at the Khong-Chiam station





is reproduced by all the precipitation products with a CC of greater than 0.91, with MSWEP and TERRA showing
the lowest bias and RMSE. ERA5 and CHIRPS performed well at station Missao-Icana in the Amazon with a CC
of 0.9 and RMSE of about 610 m3/s. For stations Savannah, Mirani-Weir, and Dunaalmas, MSWEP is the best
product with higher CC (> 0.72) and KGE (> 0.62) and lower Pbias and RMSE (Figures 5d-5f).
Table 2. KGE of monthly predictions for selected stations in basins of Africa (Niger), Asia (Mekong), South
America (Amazon), North America (Mississippi), Australia (North East Coast), and Europe (Danube).

| Basin | Stations name | Longitude | Latitude | Catchment area (km²) | ERA5 | CHIRPS | MSWEP | TERRA | CPCU | PCCSCDR |
|---|---|---|---|---|---|---|---|---|---|---|
| Niger | Lokoja | 6.8 | 7.8 | 1670000 | 0.21 | -0.1 | 0.60 | 0.34 | 0.62 | -0.99 |
| Mekong | Khong Chiam | 105.5 | 15.3 | 419000 | 0.13 | 0.56 | 0.70 | 0.91 | 0.70 | -0.04 |
| Amazon | Missao Icana | -67.6 | 1.1 | 22282 | 0.71 | 0.78 | 0.73 | 0.72 | 0.61 | 0.65 |
| Mississippi | Savannah | -88.3 | 35.2 | 85833 | 0.59 | 0.65 | 0.67 | 0.66 | 0.53 | 0.66 |
| North East Coast | Mirani-Weir | 148.8 | -21.2 | 1211 | -0.1 | 0.38 | 0.62 | 0.44 | 0.46 | -0.05 |
| Danube | Dunaalmas | 18.3 | 47.7 | 171720 | 0.34 | 0.73 | 0.78 | 0.52 | 0.71 | -0.49 |

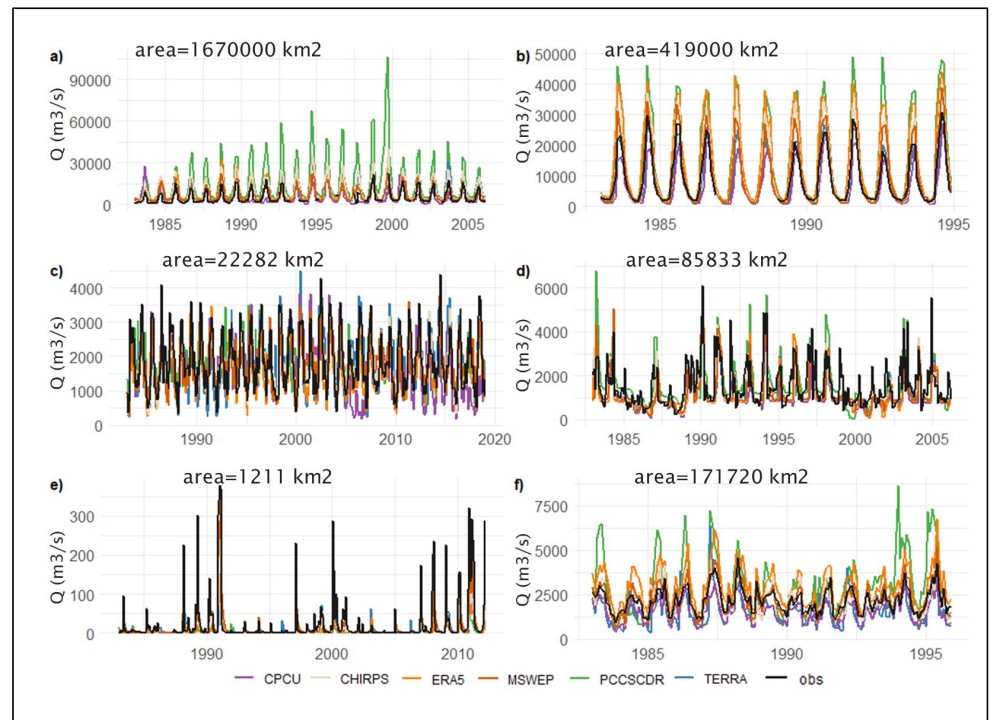


**Figure 6: Time series of monthly observed (Obs) and modelled streamflow (Q; m3/s) based on MSWEP, ERA5,**
**CHIRPS, CPCU, TERRA, and PERCCDR precipitation datasets for locations in river basins of a) Niger (Lokoja), b)**
**Mekong (Khong-Chiam), c) Amazon (Missao-Icana), d) Mississippi (Savannah), e) North East Coast (Mirani-Weir),**
**and f) Danube (Dunaalmas).**

**3.3. Performance of the precipitation datasets for daily and daily extreme discharge predictions**

Based on the daily evaluation, MSWEP followed by ERA5 show a higher CC in more than 50% of the stations
with median values of 0.41 and 0.39, respectively (Figure 7). ERA5 and MSWEP performed well in 31% and
31% of the stations with high KGE values (Figure 8). Similar to the monthly evaluation, PERCCDR shows poorer
performance (lower CC and KGE, higher biases and errors) in almost 95% of the stations. Even though ERA5
showed a higher CC and KGE in 30% of the stations it shows a higher NRMSE (up to 250%) and Pbias (up to
100%) in 20% and 30% of the stations (Figures S6 and S7). Overall, MSWEP and CHIRPS showed lower NRMSE
and Pbias compared to the other products. The CC and KGE of all the products (except CHIRPS) are lower in
North America compared to stations in South America, Europe, and Australia.

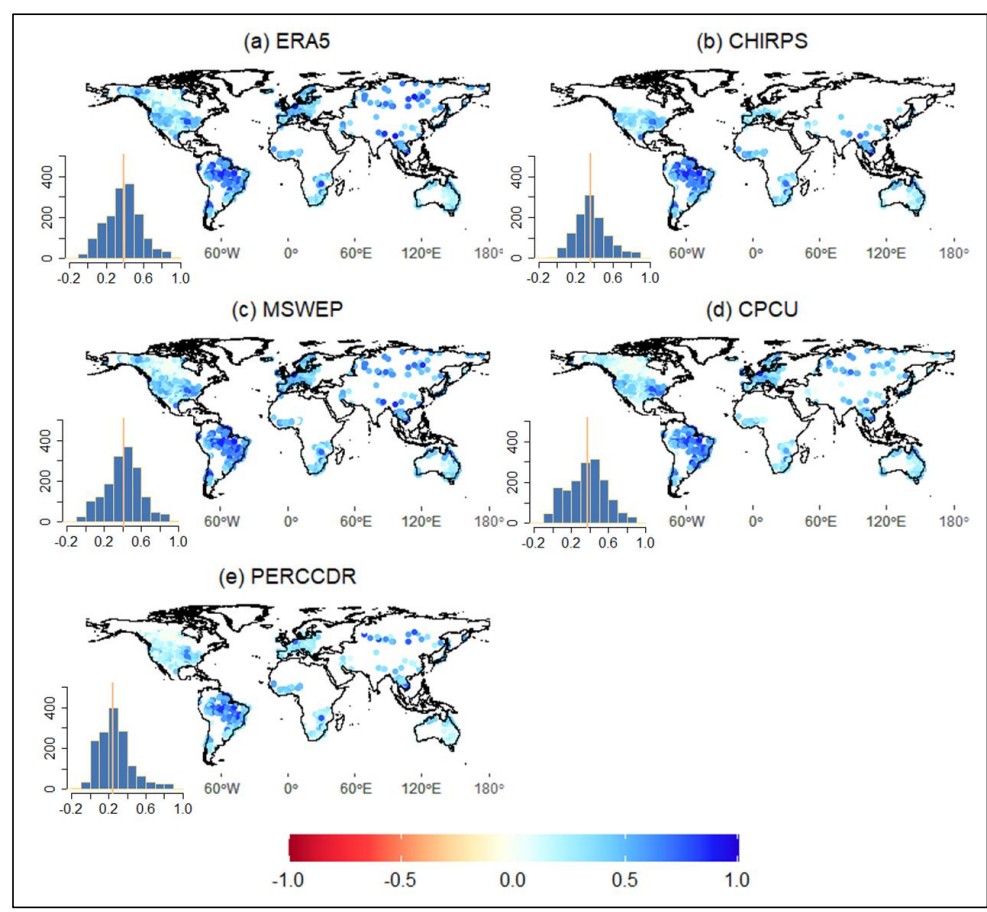

**Figure 7: Correlation (CC) between daily observed and modelled streamflow data using a) ERA5, b) CHIRPS, c) MSWEP, d) CPCU and e) PERCCDR precipitation datasets. The inset histograms show the frequency distribution of the daily CC, the yellow vertical line indicating the median value.**

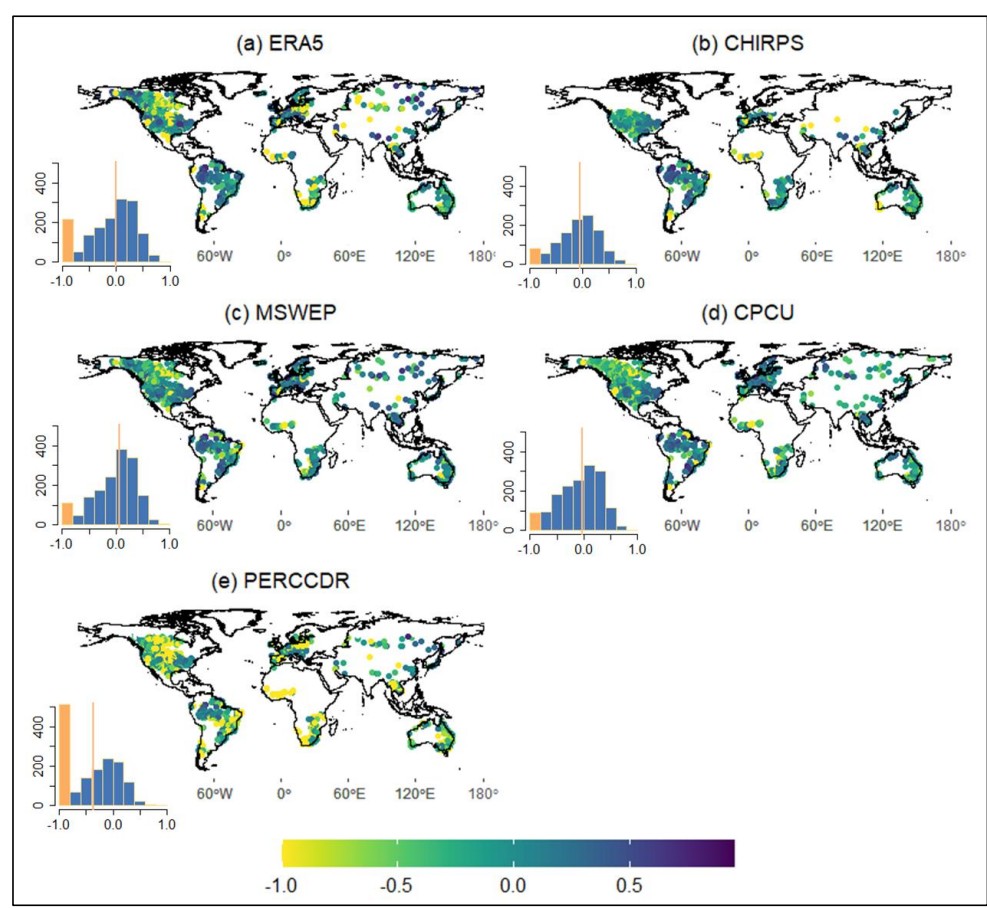

330

**Figure 8: Daily KGE values between observed and modelled streamflow based on a) ERA5, b) CHIRPS, c) MSWEP,**
**d) CPCU, and e) PERCCDR precipitation datasets. KGE values below -0.41 indicate bad model performance than**
**using observed discharge mean as a predictor. The inset histograms show the frequency distribution of the daily KGE.**
**KGE values lower than -1 are highlighted in yellow. The yellow vertical line indicates the median value.**

The performance of the daily precipitation products is also assessed for daily extremes in terms of the Q10 and
Q90 values. Based on the CC, MSWEP is the best-performing dataset for Q90 (Figure S8) and Q10 (Figure S9).
For Q10, MSWEP and CPCU exhibited a higher CC than other datasets at 38% and 32% of the stations,
respectively. Similarly, for Q90, MSWEP and ERA demonstrated a higher CC compared to other datasets at 35%
and 30% of the stations. The median CC for Q10 (Q90) is 0.32 (0.41), 0.28 (0.36), 0.27 (0.35), 0.26 (0.38), and
0.16 (0.23) for MSWEP, CPCU, CHIRPS, ERA5, CHIRPS, and PERCCDR, respectively. Similar to the annual,
monthly and daily evaluations, PERCCDR showed poor performance for the two extremes (Q90 and Q10).
Overall, the performance of the datasets is lower for extremes compared to the annual, monthly and daily scales.
Figure 9 displays differences between the observed and modelled Q10 for selected stations of Lokoja, Khong-
Chiam, Missao-Icana, Savannah, Mirani-Weir, and Dunaalmas (Table 2). Compared to ERA5, CPCU, and
PERCCDR, MSWEP followed by CHIRPS showed a higher CC (0.21-0.62) and lower Pbias and RMSE in all



stations. At stations Missao-Icana, Savannah, and Mirani-Weir, the observed Q10 is underestimated with a Pbias
of between -20% to -80%. In all the stations, the positive and negative bias is large in PERCCDR and CPCU
datasets.

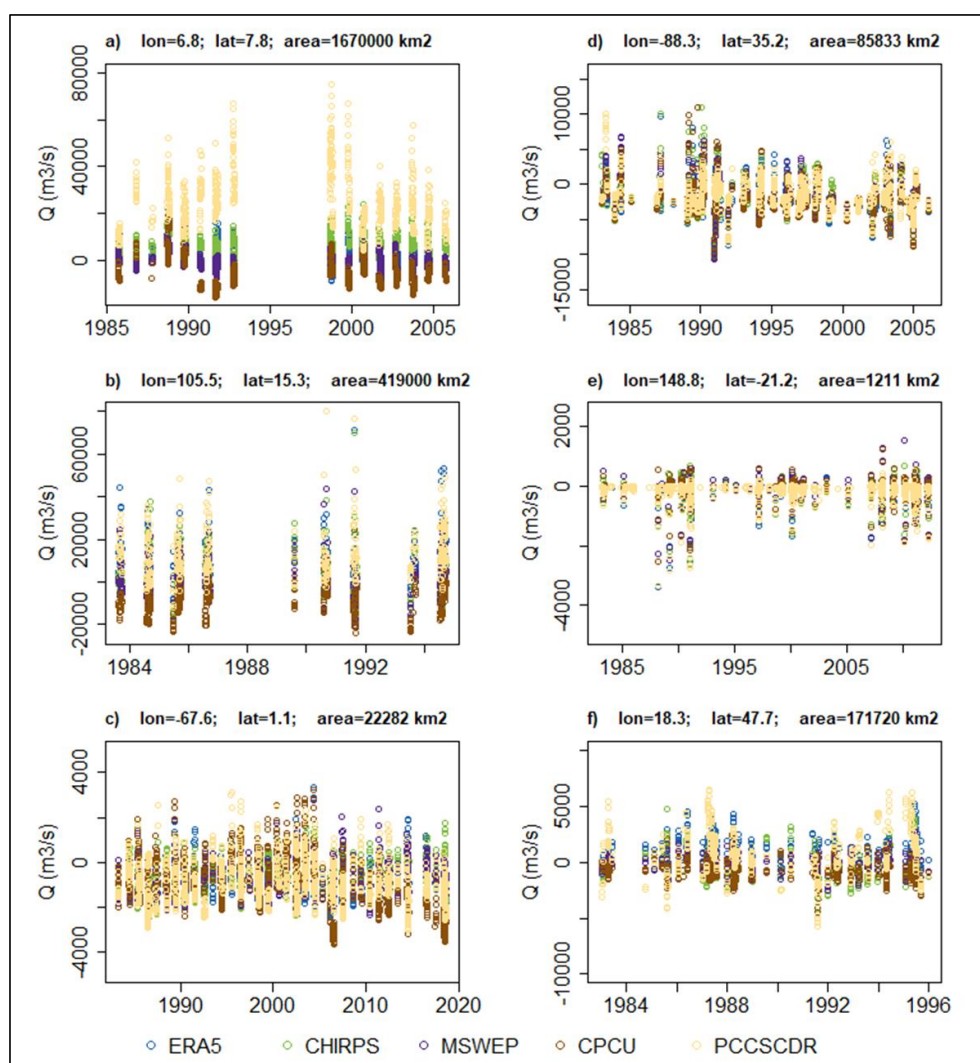


**Figure 9: The difference in Q10 (high flow) between observed and modelled streamflow based MSWEP, CHIRPS,**
**ERA5, CPCU, and PERCCDR at selected locations (Table 2) in river basins of a) Niger (Lokoja), b) Mekong (Khong-**
**Chiam), c) Amazon (Missao-Icana), d) Mississippi (Savannah), e) North East Coast (Mirani-Weir), and f) Danube**
**(Dunaalmas).**



## 4. Discussion and Conclusion

Given the challenges in representing precipitation at global scales, satellite, climate model, and reanalysis-based precipitation datasets can form the basis for monitoring and prediction of water resources and hydrological extremes, particularly in data-scarce regions of the world (Sheffield et al., 2018; Dembélé et al., 2020). Nevertheless, uncertainties and errors in these datasets require careful analysis to assess their suitability for a specific use. Error in satellite-based precipitation estimates can be due to errors in the sensor measurements, the frequency of sampling, and the retrieval algorithms, including the representation of cloud physics (Dembélé et al., 2020; Laiti et al., 2018; Alazzy et al., 2017). Climate model-based datasets, including reanalyses, have large uncertainty due to their coarse spatial resolution and ambiguity associated with model parameters (Gebrechorkos et al., 2018; AL-Falahi et al., 2020; Dembélé et al., 2020; Her et al., 2019). Reanalysis datasets may correct for some of these errors via the assimilation of observational data, but this comes with its own uncertainties due to the error characteristics of the assimilated observations and the assimilation scheme (Sheffield et al., 2006; Parker, 2016). In hydrological modelling, errors and biases in precipitation data result in poor representation of the hydrological responses and affect applications (Maggioni and Massari, 2018; Zambrano-Bigiarini et al., 2016). For example, according to Bárdossy et al. (2022), uncertainty in precipitation can lead to hydrological model errors of up to 50%. Hence, it is important to assess the quality and accuracy of the precipitation products before using them in global or basin-scale hydrological models. In data-limited regions, hydrological models driven by precipitation datasets developed from satellite sources, reanalysis or climate models are the only plausible way to represent the terrestrial water cycle (van Huijgevoort et al., 2013).

In light of the above, this study assesses the performance of selected global and quasi-global precipitation datasets for global hydrological modelling. It is important to note that this study assesses the precipitation datasets without calibration of the WBMsed model for each dataset, which could theoretically improve their performance in replicating observed river discharge. Within this context, our objective is not to evaluate the absolute performance of the hydrological model, which can be influenced by local factors, rather our focus is on comparing the relative performance of these datasets at individual locations across various precipitation datasets. Based on the evaluation at annual, monthly and daily time scales and analysis of daily extremes, no single precipitation dataset consistently exhibits high accuracy across all geographical regions, nor is one consistently better than the other datasets. This finding is in line with previous studies (Beck et al., 2017a; Dembélé et al., 2020). A similar pattern of varied performance (e.g., lower in Africa and the central United States and better in Europe) by different global hydrological models and precipitation datasets has been presented (Beck et al., 2017a; Lin et al., 2019; Harrigan et al., 2020). In addition to the uncertainty in the precipitation datasets, the poorer performance in some regions presented in this and previous studies (Beck et al., 2017a; Lin et al., 2019; Harrigan et al., 2020) can be due to the lack of representation in the hydrological models of anthropogenic influences, such as for agriculture, irrigation, water supply, and energy production.

Comparably, MSWEP and ERA5 consistently exhibited higher CC and KGE values at over 50% of the stations across annual, monthly, and daily time scales. According to Gu et al. (2023), satellite- and reanalysis-based precipitation datasets, such as MSWEP and ERA5, can provide satisfactory performance for simulating discharge globally. The higher performance of MSWEP indicates the advantage of incorporating a large number of daily





observations from field-based meteorological stations, in addition to a large set of satellite and reanalysis datasets
(Beck et al., 2017a, 2019a). Other studies have also shown the good performance of MSWEP for hydrological
modelling in different parts of the world (Beck et al., 2017a; Lakew, 2020; Li et al., 2022a; Reis et al., 2022; Gu
et al., 2023; López López et al., 2017; Satgé et al., 2019; Ibrahim et al., 2022). For example, Satgé et al. (2019)
evaluated 12 satellite-based precipitation estimates such as MSWEP, CHIRPS and PERSIANN-CDR in South
America (Lake Titicaca region) and found MSWEP was the best precipitation dataset for realistic simulation of
river discharge. MSWEP was also found to be the most reliable precipitation dataset compared to multiple datasets
such as CHIRPS and CMORPH for hydrological and climate studies in basins of Eastern China (Shaowei et al.,
2022; Wu et al., 2018). Figure 10 displays the datasets with higher CC and KGE values for modelling daily
discharge, offering a clear depiction of the spatial variability in precipitation dataset performance.

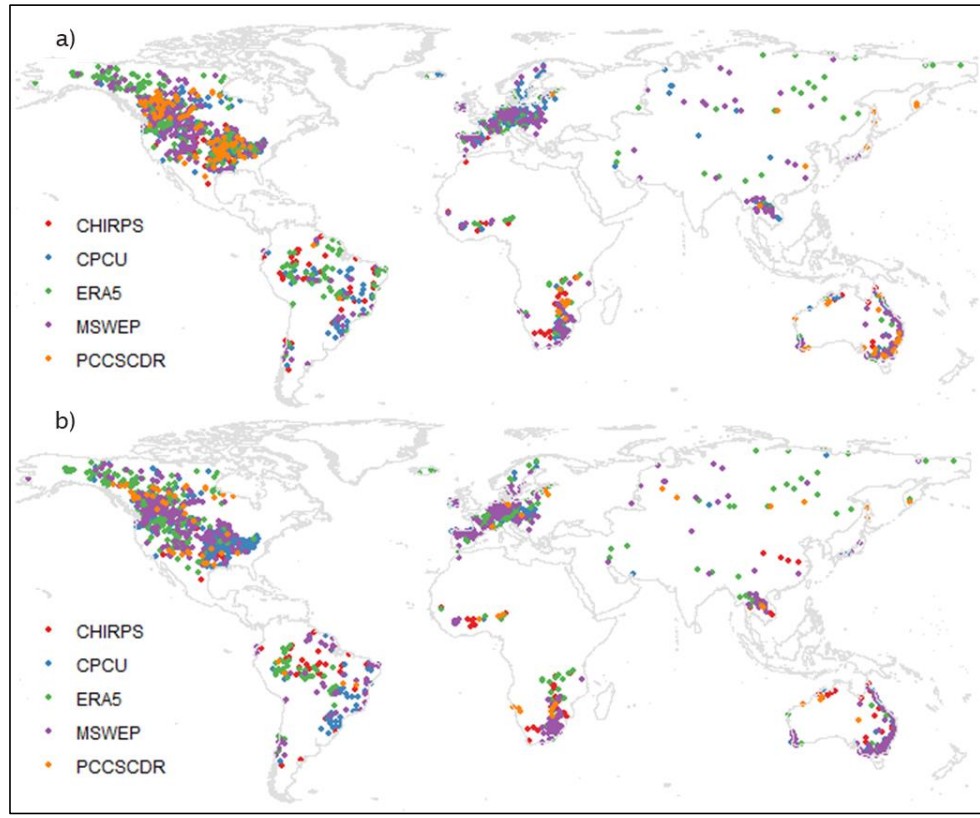


**Figure 10: The best performing precipitation dataset (CHIRPS, CPCU, ERA5, MSWEP, and PERCCDR) at each of**

**the observed discharge stations based on daily CC (a) and KGE (b).**

Even though ERA5 showed a higher KGE and CC than MSWEP, CHIRPS and TERRA in about 32% of the
stations it showed a higher error and biases. Previous studies have revealed bias and errors in ERA5 precipitation
(Lavers et al., 2021; Bechtold et al., 2020; AL-Falahi et al., 2020; Jiang et al., 2023; Lavers et al., 2022), which
leads to propagated errors and bias in hydrological modelling outputs. Harrigan et al. (2020) also reported large



biases in ERA5-driven hydrological simulations in the Central United States, South America (e.g., Brazil), and Africa. According to Lavers et al. (2022), ERA5 precipitation is more reliable in extratropical areas compared to tropical areas. Despite CPCU being a gauge-based precipitation dataset it did not show as good performance as MSWEP and ERA5 on annual, monthly, and daily timescales. In addition to the lower KGE and CC, CPCU showed higher bias and error, particularly on annual and monthly time scales. The bias and errors in CPCU can be due to the coarse resolution (0.5°) and the limited number of stations used to develop the datasets, particularly in Africa and South America. According to Beck et al. (2017a), CPCU can be used in large river basins with dense meteorological stations but can be disadvantageous in Africa and South America. This highlights the need to expand and maintain the meteorological stations in these regions, but also the need to draw from satellite and model data sources. The PERSIANN-CDR is the least-performing product with lower KGE and higher errors and biases, which has been highlighted elsewhere in terms of its inability to represent precipitation extremes (Miao et al., 2015; Solakian et al., 2020).

The precipitation datasets show limited skill overall in reproducing daily extremes (high and low flows), relative to the annual and monthly time scales. MSWEP and CPCU have shown a high CC in about 38% of the stations. This is consistent with the findings of Tang et al., (2019) for the Mekong River Basin. CHIRPS and PERSIANN-CDR are the least skilful in capturing extremes with a very low CC and large positive and negative biases (Araujo Palharini et al., 2021). For instance, numerous precipitation products have been observed to both underestimate and overestimate low and high precipitation values in Brazil (Palharini et al., 2020), consequently resulting in corresponding underestimations and overestimations of low and high streamflows. In general, several studies have concluded that precipitation datasets exhibit a substantial disparity in daily extreme precipitation events (e.g., Araujo Palharini et al., 2021; Jiang et al., 2019; Huang et al., 2022), which can be attributed to factors such as inaccuracies in satellite sensors, retrieval algorithms, temporal sampling, and satellite-observation merging and bias correction procedures used, particularly in gauge-limited regions (Miao et al., 2015; El Kenawy et al., 2015; Shen et al., 2010; Jiang et al., 2019). In addition to the uncertainty of the precipitation datasets, the limited availability of hydrological observations limits the ability to assess these datasets globally, especially for extreme flood and drought events (Brunner et al., 2021).

Overall, the evaluation presented in this paper underlines the importance of selecting high-quality precipitation datasets to drive hydrological models. Since no single precipitation dataset was found to be adequately accurate everywhere, this study can help identify the best precipitation products for any basin or region under consideration. Based on our results, MSWEP is the best overall choice but there are regions where ERA5, CHIRPS and CPCU were better overall (e.g., see Figure 10). All the precipitation datasets, particularly ERA5 and PERCCDR, require bias correction before being used to drive hydrological models in regions like North America, Asia, Africa, and Australia. For data-scarce regions such as Africa and Asia, it is difficult to recommend a precipitation dataset due to the limited number of hydrological stations used in this study. Finally, improving the precipitation datasets by adding more ground observations, for example, and by better representing anthropogenic drivers in hydrological models has the potential of considerably improving global and regional hydrological predictions.





**Data availability**

The selected precipitation datasets used in this study are openly accessible to the public. ERA5 is freely available from the Copernicus Climate Data Store (CDS; https://cds.climate.copernicus.eu/cdsapp#!/dataset/reanalysis-era5-land?tab=overview). CHIRPS can be obtained from the Climate Hazards Group (CHG; https://www.chc.ucsb.edu/data/chirps/). Access to the MSWEP precipitation dataset is provided through the GloH2O website (https://www.gloh2o.org/mswep/). TERRA is accessible from the Climatology Lab website (https://www.climatologylab.org/). CPCU is publicly available through the NOAA Physical Sciences Laboratory (PSL; https://downloads.psl.noaa.gov/Datasets/cpc_global_precip/), and PERCCDR can be freely accessed through the Center for Hydrometeorology and Remote Sensing (CHRS; https://chrsdata.eng.uci.edu/).

**Author contribution**

SG, JL, and SJD conceived the study, incorporating input from all co-authors. SG led the global hydrological modelling, while JL, SJD, and LS assisted with data management and computational resources. SG was responsible for evaluating various precipitation datasets for hydrological modelling and drafted the initial manuscript. SC provided the hydrological model and input parameters. MW, GB, RB, PD, HG, EV, YL, RH, LH, SM, and JN executed extensive data quality control and identified stations for evaluation. PA, HC, AN, AT, and JS provided code, methods, and guidance. DP, SJD, and SED supervised the research and secured funding. All authors contributed to investigating research findings and played integral roles in manuscript writing and editing.

**Competing interests**

We declare that Louise Slater is a topical editor of Hydrology and Earth System Sciences (HESS).

**Acknowledgements**

This work is part of the Evolution of Global Flood Hazard and Risk (EVOFLOOD) project [NE/S015817/1] supported by the Natural Environment Research Council (NERC), the UK Foreign, Commonwealth and Development Office (FCDO) for the benefit of developing countries (Programme Code 201880) and the UK's Natural Environment Research Council (NERC; NE/S017380/1).



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
