# Peer review of "Global scale evaluation of precipitation datasets for"

_Hydrology and Earth System Sciences, 2023_

## Author Response (AR1)

**Reply to Referee #1**

We appreciate the reviewer for dedicating their time and providing valuable, detailed feedback. We have diligently addressed all of the comments and discussed them as follows.

Review comments

Thank you, HESS, for inviting me to review this paper. I like the paper as it covers the global scale analysis of various reliable precipitation datasets using a global hydrological model. Selecting the best-performing global dataset for precipitation is meaningful for areas where observation data is scarce.

Authors´ response:

- Thank you for recognizing our efforts and providing feedback to improve the quality of the paper.

Comments

1. There are no keywords provided in this manuscript.

Authors´ response:

- Thank you for the suggestion. However, we believe it may not be advised in HESS. We will discuss this with the editorial team, and if deemed appropriate, we will add keywords to the manuscript.

2. What are the values of the x and y-axis of the inset histograms? Mention them in the caption at least.

Authors´ response:

- Thank you for bringing this to our attention. We appreciate the clarification. The inset histograms illustrate the frequency distribution of the CC and KGE values for each station in each simulation based on different precipitation datasets. To enhance clarity, we will include additional information in the text and figure legend, specifying that the y-axis represents frequency, while the x-axis represents CC and KGE values.

3. Long sentences in lines 60-66, 87-92, 97-101, 123-127, 196-201, 220-222

Authors´ response:

- Thank you for suggestion. Based on your recommendation, we have shortened the long sentences for clarity.

4. Line 203, write table 1 as Table 1

Authors´ response:

- Thank you. table 1 is not modified as Table 1.

5. Line 208, remove or take this "Cohen et al. (2022) report $R^2$=0.99 in 30-year average prediction against USGS gauge data and a global river dataset." sentence in the discussion section

Authors´ response:

- Thank you! This sentences is now removed.

6. In Figures 3 and 10, you have provided the best-performing precipitation dataset based on annual CC and KGE. However, I can't see the values associated with CC and KGE in the figure; it shows the global distribution of the dataset.

Authors´ response:

- The figures illustrate the global distribution of various precipitation datasets based on their highest values of CC and KGE. For clarity on the specific values associated with CC and KGE, please refer to Figure 1 (for annual CC), Figure 2 (for annual KGE), Figure 7 (for daily CC), and Figure 8 (for daily KGE). These figures provide detailed information on the corresponding CC and KGE values for a comprehensive understanding of the dataset performance.

7. In Figures 2 and 5, the authors said KGE values lower than -1 are highlighted in yellow. But as I can see the x-axis it seems different.

Authors´ response:

- Thank you for pointing out this. The x-axis values are adjusted to indicate the centre of the bar. The vertical red line represents the median values, while values lower than -1 were highlighted with an orange bar.

8. For Figures 6 and 9, please make a superscript of the units for discharge and area inside the figures and other parts of the manuscript.

Authors´ response:

- Thank you for your feedback. We have revised the figures and other parts of the manuscript to include superscripts for the units of discharge and area in Figures 6 and 9.

**Reply to Referee #2**

We appreciate the reviewer for dedicating their time and providing valuable, detailed feedback. We have diligently addressed all of the comments and discussed them as follows.

In the submitted manuscript, the authors assess the efficacy of six global/near-global precipitation datasets in streamflow modeling across 1825 gauging stations, employing the WBMsed hydrological model. Building upon prior research, this investigation scrutinises precipitation datasets that have either been updated or not been included in previous studies. Notably, the assessment extends beyond the daily time scale, including monthly and annual perspectives, and evaluates the model's proficiency in simulating both high and low flows for each of the selected precipitation sets.

From a linguistic perspective, the manuscript is well-written and exhibits a clear structure. It falls comfortably within the scope of the journal and will capture the interest of members within the hydrological modeling community. However, it is imperative to acknowledge certain significant shortcomings which need to be addressed before the manuscript can be considered for publication.

Authors´ response:

- Thank you for recognizing our efforts and providing feedback to improve the quality of the paper. **For the following general comments, we have provided a detailed answer in the "Specific comments" section.**

1. The methods section is comparatively short with a main focus on the description of the precipitation sets evaluated and it does not provide sufficient detail for reproduction of the study.
   1. Eleven of the authors claim their main contribution to this study was extensive data quality control, however, I cannot find a single sentence mentioning or describing the quality control procedure applied in the methods section.

Authors´ response:

- Thank you for the feedback. We have added more information to the methods section to facilitate the reproduction of the work, including details on the quality control procedure. **Additional information is provided in the 'Specific comments' section.**

   2. It is unclear which datasets were used to drive the model as all that is given are two references to previous papers and a rather confusing description of some update applied.

Authors´ response:

- Thank you. We have added a table listing the datasets used to drive the model, in addition to the precipitation datasets, for clarity. **Additional information regarding this is provided in the 'Specific comments' section.**

3. The reader is left guessing as how the mismatch between resolution of the precipitation data and the resolution of the hydrological model was addressed.

Authors´ response:

- Thank you. We have addressed the concern by adding a section that explains how the datasets are pre-processed to the same resolution to run the hydrological model at 0.1°. **More details are available in the 'Specific comments' section.**

4. No rationale is given for how and why the 1825 gauging stations were selected for evaluation. Furthermore, the authors do not provide a list containing the GRDC number and/or name of the selected gauging stations, which makes reproduction of this list impossible (maybe this is planned to be provided in electronic form at a later stage?).

Authors´ response:

- Thank you for bringing this to our attention. The selection of gauging stations is based on the observed length records to evaluate the modelled discharge. The stations' information, including lat, lon and area, is available from GRDC (https://portal.grdc.bafg.de/applications/public.html?publicuser=PublicUser#dataDownload/StationCatalogue), and we are willing to provide the locations if necessary. **More details are available in the 'Specific comments' section.**

2. According to the abstract, the authors want to provide guidance on the selection of precipitation sets for hydrological modelling. They also make an enormous effort and use five different metrics for performance evaluation. Sadly, the manuscript does not provide much beyond the pure quantitative evaluation. Giving a rationale for the selection of those specific metrics and discussing why one/some of them provide a benefit for the evaluation under certain conditions would tremendously improve the manuscript. It would also be of value if the manuscript could discuss the performance within specific climatic zones (e.g tropics vs temperate) as suggested in the abstract.

Authors´ response:

- Thank you for the suggestion. We have added a section highlighting the performance of the different precipitation datasets in different regions of the world by selecting large basins in South America (Amazon), Europe (Danube), USA (Mississippi), and Africa (Orange), We have also added a few lines describing the benefit of the selected statistical methods in methodology section (section 2.4) for clarity. **More information is available in the 'Specific comments' section.**

3. The discussion section would benefit from further editing. It contains paragraphs/sentences which would be better placed in other sections (see specific comments below) and it does not discuss any limitations of the study.

Authors´ response:

Thank you for providing the edits and suggestions in the discussion section. Based on your recommendations and edits, we have modified the section. We have also added a

section about the limitations and uncertainities of the study. **Additional information is provided in the 'Specific comments' section.**

4. Presenting maps at a global scale is no easy task and a lot of effort has been put into a substantial number of figures. However, it is very difficult to actually see what is going on. The coloured points often overlap each other and it is not easy to distinguish the colours or to discern the differences between the precipitation sets. Providing maps limited to regions which exhibit major differences between the sets might be a better option. There also seems a consistent problem within the caption where 'yellow' lines or highlights are referred to but the presented colour seems to be more 'orange' or 'red'. Replacing the coloured line with a dashed black line and highlighting the bars in grey would avoid any potential colour confusion. Furthermore, the colour legends seems to miss a title describing what value is actually presented.

Authors´ response:

- Thank you for the suggestions and for understanding the challenge of presenting global scale evaluation results. To address your comment, as discussed above, we have now provided detailed maps highlighting the performance differences of various precipitation sets by selecting large river basins. We believe this improvement enhances result visibility. Additionally, we have adjusted the colours in each figure and added a colour legend for clarity. **Further details on these modifications can be found in the responses to specific questions.**

**Specific comments:**

Line 30: The term 'more than 1800' is rather vague and should be changed to the exact number of stations.

Authors´ response:

- Thank you. We have changed this to the exact number of stations (i.e., 1825).

Lines 34-37: The sentence seems overly complex. Removing the vague quantification and replacing it by the information in the brackets should address the issue.

Authors´ response:

- Thank you for your suggestion. We have revised the sentence in lines 34-37 as recommended.

Line 88: Again, the expression 'multiple' is rather vague and should be changed to the exact number of sets evaluated.

Authors´ response:

- Thank you. We have modified this by adding the numbers, which are 22 precipitation datasets.

Line 88-91: Listing all those example sets bloats the paragraph. I recommend to remove the list of example sets in the brackets.

Authors´ response:

- Thank you for your suggestion. We have removed the examples as recommended.

Line 92-93: Same as in line 88-91

Authors´ response:

- Thank you for your suggestion. We have removed the examples as recommended.

Lines 86-105: The authors mention Voisin et al. (2008) but do not give any information on the scope of this cited study or how it relates to the presented work.

Authors´ response:

- Thank you for bringing this to our attention. Voisin et al. (2008) conducted a global-scale evaluation of precipitation (ERA-40 and GPCP) for hydrological modelling, providing relevant context for our study. We have added a line to explain this in the manuscript.

Line 121: I suggest changing the header to 'Precipitation sets'

Authors´ response:

- Thank you for the suggestion. We have changed the header as recommended ("Precipitation datasets").

Line 122: What is the rationale behind choosing datasets of length >30 years? Why those specific six sets?

Authors´ response:

- As we explained in the introduction, we have tried to build on the work by Beck et al., (2017a). Based on Beck et al., (2017a), we excluded the datasets with very low performance and selected datasets with the highest performance, such as older versions MSWEP and CHIPRS. We then added additional datasets that were not evaluated globally before. The length requirement stems from our interest in examining long-term changes in river flows, including droughts and floods, globally. In line with the length criterion, we have added a few lines explaining why we chose datasets greater than 30 years.

Line 130-136: It is not entirely clear which version of the set has been used. Presumably ERA5-Land? Why was it selected and was it used/evaluated in previous studies?

Authors´ response:

- It is correct. We have used ERA5-Land due to the improvements compared to ERA5, including the spatial resolution, and because it was not evaluated globally for hydrological modelling together with other datasets such as MSWEP and CHIRPS. To make this clearer, we have edited the section by adding such as: "*In addition, ERA5-Land, a subset of ERA5 focusing on land areas, delivers more detailed climate information at higher spatial resolution (0.1°) from 1950 to the present compared to ERA5 (Hersbach et al., 2020). Here, ERA5-Land (referred to as ERA5) is used to evaluate its performance for global hydrological modelling.*"

Line 137-147: Which version has been used and at what resolution? Why was CHIRPS selected for evaluation?

Authors´ response:

- Thank you for bringing this to our attention. We utilized version 2.0 of CHIRPS with a spatial resolution of 0.05°. The selection of CHIRPS was based on its continued development and widespread use in various studies. Additionally, its performance, as recommended in a previous study by Beck et al. (2017a), demonstrated effectiveness across different basins worldwide.

Line 148-161: The description of the set is very detailed and I recommend to re-consider listing all the different data sets which were combined to create MSWEP. Instead you could provide information on which temporal resolution was selected and why this set was chosen for evaluation.

Authors´ response:

- Thank you for your suggestion. We have revised the section by providing detailed information on the spatial and temporal resolution of MSWEP and explaining the reasons for its selection in the evaluation.

Line 162-168: Why was TERRAClimate selected for evaluation? Has it been used in other studies?

Authors´ response:

- Thank you for your inquiry. TERRA Climate was included in the study due to its very high resolution (~4 km), making it suitable for detailed hydrological modelling. It had not been previously compared with other datasets for hydrological modelling and its performance in a global context was not well-known. Despite not being as widely used as ERA5, MSWEP, and CHIRPS, its exceptional spatial resolution made it a potential candidate for evaluation in this study.

Line 169-174: Why was CPCU selected, what is the temporal and spatial resolution of this set? Has it been used/evaluated in previous studies?

Authors´ response:

- CPCU was chosen for evaluation because it relies on ground observations. Despite the limited global distribution of observed datasets, there was a curiosity to assess how a dataset based solely on ground observations would perform in comparison to datasets that incorporate both observations and satellite data. CPCU has a relatively coarse resolution (0.5°) compared to other datasets. It had been previously evaluated by Beck et al. (2017a) on a daily time scale, and in this study, an examination was conducted to see how CPCU performs on monthly and annual time scales, as well as in extreme conditions.

Lines 121-184 Section 2.1: The section provides a description of the sets (some more extensive than others) but does not provide any information on how the sets were prepared for use in the hydrological model. As the model was run at daily time steps and 0.1° resolution I assume that most of the sets were interpolated to the model's resolution?. Furthermore, there is no information provided on how the monthly data in TERRA was converted to daily data.

Authors´ response:

- Thank you for bringing this to our attention. The datasets were indeed interpolated to a resolution of 0.1° within the hydrological model. **We have added a brief explanation on how the datasets were prepared for the hydrological model, including details on the interpolation method used in section 2.2**. as: "*All the input precipitation datasets are bilinearly interpolated to the same spatial resolution of 0.1°.* "
- As described in section 2.2 (*Even though WBMsed can disaggregate monthly time series into daily, TERRA (only available at monthly resolution, see Table 1) is evaluated on monthly and annual time scales, whilst all other datasets are evaluated at daily, monthly and annual time scales*), TERRA is not converted into daily. Hence, the model is run at monthly and daily time scales.
- The hydrological model can implement temporal downscaling (monthly to daily) using input daily rainfall characteristics from other datasets. Although this is a common practice in hydrological modelling, especially in data-sparse regions, it does not replace the original data. Therefore, we have decided not to evaluate TERRA on a daily basis and for daily extremes

Line 194: What does high-resolution mean in this context? Is it finer, equal or coarser than the 0.1° resolution at which the model was run?

Authors´ response:

- Thank you. The resolution of the river network we used is 0.1°. For clarity, we have removed the term "high resolution" as it is explained in detail in the next section: "*6 arc-minute HydroSTN30 network derived from HydroSHEDS (Lehner et al., 2008)*"

Lines 196-201: It is unclear which sets have been used to drive the model. Please, provide a table listing all the data sets and where to obtain them. Has the model been calibrated? Which parameter sets were used?

Authors´ response:

- Thank you for the suggestion. We have added a table (Table S1) listing the additional datasets used to drive the model, in addition to the precipitation sets, instead of referring to the paper (Cohen et al., 2013). As we tried to explain in section 2.2, the model is calibrated globally and shows an $R^2$ of 0.99 for the long-term average (Cohen et al., 2022). This part is now moved to section 4 as recommended. We have now included a line explaining the additional parameters used in the model in section 2.2.

Lines 202-204: It is mentioned that the option to disaggregate TERRA from monthly to daily time steps was not used. If that is the case, I wonder how the model could be run at daily time steps. Was another method used for disaggregation?

Authors´ response:

- As explained earlier, the model can run on both daily and monthly time steps depending on data availability. Therefore, we ran the model using both daily and monthly time steps. We have included a line to clarify this in the revised version.

Lines 205: The model was run at 0.1° resolution but several of the evaluated precipitation sets have a different resolution. How was this mismatch addressed? Did you apply any interpolation?

Authors´ response:

- We hope this is clear now. As mentioned earlier, yes, we interpolated all the datasets to 0.1° within the model. A brief explanation has been added in section 2.2 for clarity as *"All the input precipitation datasets are bilinearly interpolated to the same spatial resolution of 0.1°."*

Lines 208-209: The last sentence in this section (Cohen et al. (2022) report … ) should be removed. It could be used in the discussion section.

Authors´ response:

- Thank you for the suggestion. We have removed and moved it to the discussion section as recommended.

Lines 211-218: How were the stations selected? Were stations limited to those with a record of > 10 years ? Were gaps allowed in the record? Will a list of the stations be provided? As 11 authors claim to have performed extensive quality control, at least one full paragraph outlining the steps involved should be provided.

Authors´ response:

- Thank you for the suggestion. Yes, we only consider stations with at least 10 years of record and gaps were allowed in the record. Processing the datasets for the evaluation was a time-consuming task, and we decided to split the jobs between the co-authors. If a list of stations is required, we are happy to supply it at any time but they are also available on GRDC with full details.

- As per your recommendation, we have now added lines explaining the quality control, including the criteria for station selection based on length and data gaps as "*Here, we consider stations with a minimum record length of 10 years, allowing for missing values within this period.*"

Lines 237-239: I recommend to remove these two sentences.

Authors´ response:

- Thank you for the suggestion. The sentence is now removed.

Lines 259-261: Does the figure show monthly or annual CC values in the histograms? I assume the 'yellow' line refers to the 'orange' or 'light red' line in the inset?

Authors´ response:

- Thank you for pointing this out. Figure 1 represents the annual CC. We have changed the vertical line to red and modified the text as follows: "*The inset histograms show the frequency distribution of the annual CC, with the red vertical line indicating the median value.*"

Lines 265-267: The description reads 'values lower than -1 are highlighted in yellow' but I cannot see any yellow highlights, just an 'orange' bar and an 'orange' line. Furthermore, the highlighted bars seem to refer to values between -0.8 and -1.0 rather than to those < -1.

Authors´ response:

- Thank you. We have changed this to orange and the line to red as: "*KGE values lower than -1 are highlighted in orange. The red vertical line indicates the median value*". Additionally, we modified the figure to clearly indicate the values on the axis.

Line 268 (Figure 3): I understand the reason behind presenting this map but it is really hard to see what is going on. Maybe limiting the presentation to areas in which there are clear differences between the datasets would be better?

Authors´ response:

- Thank you for the suggestion. We agree and have added additional maps to highlight specific areas with clear differences, focusing on large river basins such as South America (Amazon), Europe (Danube), USA (Mississippi), and Africa (Orange). We included a section in 3.2 and 3.3, which showcases the performance of the precipitation datasets at these basins at both monthly (Figure 6) and daily (Figure S11) time scales.

[Figure]

**Figure: Performance of precipitation datasets (ERA5, CHIRPS, MSWEP, TERRA, CPCU, and PERCCDR) at discharge stations in a) Amazon, c) Mississippi, e) Danube, and g) Orange river basins based on their monthly CC. Performance of the datasets based on KGE for the Amazon, Mississippi, Danube, and Orange River Basins is illustrated in figures b, d, f, and h, respectively.**

Lines 280-282: 'Yellow' looks actually 'orange', see above.

Authors´ response:

- Thank you. This is now changed to red and reads as. "*the red vertical line indicating the median value*".

Lines 292-296: The same issue as in Lines 265-267.

Authors´ response:

Thank you. The figure and text is now modified as "*KGE values lower than -1 are highlighted in orange, with the red vertical line indicating the median value.*"

Line 312 (Figure 6): This figure is really difficult to read. I would suggest to remove the annotation stating the catchment area as this information is already provided in Table 2 and because it distracts from the time series. Can the readability be increased by showing one single time series per row and increasing the height of the figure?

Authors´ response:

- Thank you for the recommendation. We have removed the text from the figures and increased the height. To reduce the number of figures, we move the figure to the supplementary material (Figure S5) as the information is already provded in Table2.

Line 327 (Figure 7): Yellow' looks 'orange', see above.

Authors´ response:

- Thank you. This is now modified and changed the color to red.

Line 330 (Figure 8): Yellow' looks 'orange', see above.

Authors´ response:

- Thank you. This is now modified as; "*KGE values lower than -1 are highlighted in orange, with the red vertical line indicating the median value.*".

Line 349 (Figure 9): I have a hard time understanding what is going on in this figure. Reading the description I was expecting much less cluttered plots than presented. Why are there so many points per catchment? On what time scale did you actually calculate the extremes? I suggest, you add a section in the methods detailing the calculation of your Q10/Q90 to clarify potential misunderstandings. Furthermore, I suggest to remove the annotations presenting the lat, lon and area information as this is already given in Table 2 and is unnecessary here.

Authors´ response:

- We also agree that this is very difficult to understand the variability of the performance of the individual precipitations as it shows a daily extreme above the 90[th]

(Q10, highflow) and 10th (Q90, lowflow) percentiles. We have removed this figure and instead, we added a spatial map illustrating the correlation coefficient (CC) for Q10. Additionally, we have refined the methodology of computation of Q10 and Q90 for clarity.

Line 355-372: I suggest to remove this paragraph and put it into the introduction instead to provide more background information.

Authors´ response:

- Thank you for your suggestion. We have incorporated your feedback and moved the section to the introduction.

Line 373-374: This first sentence (In light of the above … for global hydrological modelling) better suits the introduction as it outlines the aim of the study.

Authors´ response:

- Thank you. We have removed this line and included it in the introduction.

Lines 374-376: This is a crucial bit of information which I was looking for in the methodology (It is important to note … replicating observed river discharge). It should be placed into the section describing the model setup.

Authors´ response:

- Thank you. We have moved this and updated the methodology section 2.2 as recomended.

Lines 376-387: This sentence (Within this context … various precipitation sets.) would be better placed into the introduction, outlining the objectives of the study.

Authors´ response:

- Thank you for your suggestion. We have moved and updated the introduction as recommended.

Lines 379-380: This sentence (Based on the evaluation … better than other datasets) is an excellent opening sentence for the discussion section.

Authors´ response:

- Thank you for the suggestion. This line is used as the opening of the discussion section as recommended.

Line 400-401: Why is it necessary to present a new figure here? The variability in performance has already been shown in Figure 3. This last sentence should be removed.

Authors´ response:

- Thank you for the recommendation. We have removed the sentence and the figure.

Line 402 (Figure 10): I do not think that this figure provides anything that has not yet been covered by the other figures. Better place it into the supplemental material.

Authors´ response:

- We have moved Figure 10 to the supplemental material as recommended. Thank you!

**Technical corrections:**

Line 33: Remove comma after 'Whilst'

Authors´ response:

- Thank you. The comma is now removed.

Line 99: This is the first time ERA5 is mentioned in the main body and the acronym hence needs explaining. Should be changed to 'such as ECMWF Reanalysis v5 (ERA5)'

Authors´ response:

- Thank you! The revised version now includes the full name, "the fifth generation of the European Centre for Medium-Range Weather Forecasts (ECMWF) Reanalysis (ERA5)."

Line 101: Is it PERCCDR (as in the abstract and methods) or PERSIANN-CCS-CDR? Please, double check consistency of acronyms.

Authors´ response:

- You are right; it is PERCCDR. Corrected!

Line 123-127: Some of the acronyms have already been defined in Lines 99-101 and do not need to be defined again.

Authors´ response:

- Thank you! We have modified this section as recommended.

---

## Author Response (AR2)

Dear Prof. Yi He,

Thank you for handling our paper, and we have addressed your comments accordingly. Additionally, we have updated the "Author Contributions" section as per your recommendation.

Thank you for responding to the review from Referee #2. I have some additional comments as follows. I look forward to receiving your responses.

CC and KGE in the abstract need full names

Authors´ response:

- Thank you! We have added the full names of CC and KGE.

L58: missing 'OF', Whilst precipitation is one OF the most important components.
Authors´ response:

- Thank you. This is now modified.

L127: A comprehensive and physically based gridded global hydrological model (WBMsed; Cohen et al., (2013)) is used to simulate river discharge globally. Upon reading the paper by Cohen et al., (2013), I understand the WBMsed model is 'a spatially and temporally explicit (pixel-scale and daily) implementation of the BQART river-mouth sediment load model (Syvitski and Milliman, 2007)'. Please explain the reason for selecting this model in a global assessment of streamflow.

Authors´ response:

- As explained in section 2.2, WBM is one of the first global hydrological models, with comprehensive parameters and datasets including reservoirs, dams, and crop water evapotranspiration often not included in most global hydrological models. WBMsed, a combination of the latest WBM version (WBMplus) and the sediment model (BQART), enables the simulation of sediment fluxes within the WBM framework. In this paper, we indeed only analyse the hydrological predictions from the framework but opted to use the 'sed' version of the framework for consistency with our following analysis of sediment flux. The hydrological predictions by WBMsed and WBMplus are equivalent. This clarification has been added to the introduction (lines 127-128) and section 2.2 (lines 210-215) to enhance clarity.

L230: It is important to note that this study assesses the precipitation datasets without calibration of the WBMsed model for each precipitation dataset, which could theoretically improve their performance in replicating observed river discharge. The WBMsed model is a physically based model. Please explain what parameters would need to be calibrated if you were to calibrate the model.

Authors´ response:

- The parameters for calibrating the WBMsed model include soil characteristics (e.g., depth, soil moisture, field capacity, rooting depth), baseflow release time, quick flow coefficient, snowfall and snowmelt threshold, percolation fraction, maximum canopy interception storage, evapotranspiration response to soil moisture drying, speed of

wave propagation, and multiple hydraulic geometry features of the river channel such as depth, width, and velocity.

- Soil characteristics are among the most sensitive parameters often used for calibration, along with flow characteristics. For more information about the hydrological model parameters see Wisser et al., 2010.

Spatial resolution: Because the precipitation products were interpolated to the same spatial resolution of 0.1°, I wonder how this affects the model outputs. Does this favour the rainfall products that are already at 0.1° resolution. Please comment on how the spatial resolution and interpolation potentially affect the model performance and your conclusion.

Authors´ response:

- This is a very good question and may indeed have some influence, particularly when comparing to observed precipitation rather than streamflow, which represents catchments greater than 100 $km^2$ in area. If resolution were to have a significant impact, the PERCCDR and TERRA datasets, which have the highest resolution, would have performed very well. When high-resolution datasets are aggregated to a coarser resolution, they often perform well, but this is not the case for PERCCDR and TERRA. Hence, accuracy relies more on the methods and datasets used to develop the datasets rather than spatial resolution.

Reference:
- Wisser, D., Fekete, B. M., Vörösmarty, C. J., and Schumann, A. H.: Reconstructing 20th century global hydrography: a contribution to the Global Terrestrial Network- Hydrology (GTN-H), Hydrology and Earth System Sciences, 14, 1–24, https://doi.org/10.5194/hess-14-1-2010, 2010.